# The evolution of food security in Japan—Based on an indicator evaluation system including climate change indicators

**Caixia Li** *

Graduate School of Fisheries and Environmental Sciences, Nagasaki University, Nagasaki, Japan

* 15800718579@163.com

## Abstract

As climate change intensifies, food security has received widespread attention. This study examines the development of Japan's food security index and its relationship with climate change. From these findings, a food security indicator system for Japan was established. The food security system has six dimensions: availability, nutrition, climate change, society, economics, and fertilizer. The factors affecting food security are complex and variable, and this paper adds the Fertilizer Security Index (FSI) to the previous studies. The overuse of fertilizers directly contributes to soil and atmospheric pollution, which can indirectly lead to issues of food quality insecurity. Including this factor within the food security system is fully justified. This enhances the precision of the food security index system to some degree. The results showed that Japan's overall food security index exhibited a slight downward trend from 0.113 in 1980 to 0.099 in 2022. Food security reached its lowest point of about 0.067 in 1993. In addition, all five indicators, except the fertilizer index, put pressure on the food security index. Due to the uncertainty inherent in climate change, specifically its ambiguous positive and negative impacts on food security, the Climate Change Security Index provides detailed evidence in this paper supporting whether climate change contributes to or undermines food security. Finally, the study put forward recommendations to ensure food security.

## 1. Introduction

### 1.1 Background and significance

Japan has maintained a long-standing partnership with the United Nations Food and Agriculture Organization (FAO), playing a crucial role in promoting global food security and advancing the sustainable development of natural resources. However, since the late 1980s, Japanese agriculture has been shrinking annually for many reasons, such as the scarcity of arable land, low wages, falling agricultural incomes, a sharp increase in part-time and aging farmers, and a lack of farming successors [1].

With the increasing intensity of climate change and the growing frequency of extreme weather events, agriculture is one of the sectors most vulnerable to disruptions [2]. Climate

**Competing interests:** The authors have declared that no competing interests exist.

change often directly impacts agriculture through high temperatures and extreme rainfall, consequently affecting food security. Japan, an elongated island nation, exhibits significant regional differences in climate, including rainfall and temperature [3]. Over the past century, however, Japan's annual average temperature has risen by 1.30°C, leading to declines in both the quality and yield of agricultural products [4]. In 1985, excessive typhoons led to a cool summer and heavy rainfall, reducing both the yield and quality of rice [5]. In 2006, Kyushu experienced heavy rainfall, with Miyazaki Prefecture recording a maximum of 1,281 mm over seven days [6]. In 2011, Japan faced record-breaking rainfall [7].

Studies have shown that factors such as food imports and agricultural land positively influence the Food Production Index (FPI), highlighting their crucial role in safeguarding food security [8]. Nutrition is a critical factor for human well-being. A significant portion of the global population is facing the triple burden of malnutrition: the coexistence of undernutrition, micronutrient deficiency, overweight, and obesity [9]. Compared to 2019, before the pandemic, the number of people experiencing hunger increased by approximately 122 million in 2022 [10]. Japan also sees nutrition as key to sustainable development and human security. Japan was one of the first investors in the Scaling Up Nutrition movement, and today is the fourth largest contributor of ODA (Official Development Assistance) to nutrition in the world [11]. Owing to the importance of nutrition for healthy individuals, Japan has committed to engaging the private sector to utilize its strengths and technologies to improve nutrition [12]. Human-induced greenhouse gas emissions can impact nutritional health through various biophysical and socio-economic shifts [13]. Therefore, stability primarily concerns the availability and accessibility aspects of food security [14]. Carbon dioxide ($CO_2$) emissions from the food sector have steadily declined yearly [15]. Lowering $CO_2$ emissions is advantageous for combating climate change [16, 17]. A previous analysis showed that China and Japan are experiencing a phenomenon in which biofuels based on agricultural products and cellulose compete with food and food-related demand, as well as agricultural production [18], which seriously threatens food security. Therefore, adaptation to climate change has also become imperative [19].

A review of the literature reveals a lack of studies examining the relationship between food security and climate change in Japan, as well as a scarcity of systematic research on food security indicator systems. This study will examine the development of various indicators, especially those linked to climate, to assess whether they positively or negatively impact food security. The research hypothesis suggests that the Climate Change Security Index's influence on food security fluctuated between 1980 and 2022. Considering Japan's significant role in the FAO, these findings will contribute to shaping global food security strategies.

## 1.2 Literature review

The world's silence about how global climate change affects food security has resulted in hunger and malnutrition, which might worsen rapidly if developed countries neglect the issue [20]. According to food security defined by FAO [21], national food security implies that a country has sufficient food for everyone. However, this does not guarantee that all households have enough food, as distribution may be uneven [22]. Since the mid-1990s, its self-sufficiency ratio has remained at approximately 40%. This is because domestic supply (total domestic heat supply) is decreasing, whereas consumption (total heat supply) is declining [23]. In 2022, the Economist Intelligence Unit (EIU) published the Global Food Security Index Report, providing a detailed ranking of 113 countries based on four key dimensions, including the capacity to purchase food, food supply capacity, food quality and security, and natural resource resilience using a dynamic benchmark model [24]. Among them, Japan ranks 6th, which belongs

to the upstream level. Many scholars have studied food security from other perspectives or by selecting different dimensions. For instance, energy and food security are interconnected through price fluctuations, with rising oil prices negatively impacting food security [25]. Wiesmann et al. [26] examined food security based on four criteria: dietary quantity, dietary quality, and psychological, social, and cultural factors. Other studies have analyzed food security through dimensions such as availability, access, utilization, and stability, alongside five additional aspects [27]. It is widely acknowledged that sustainable food systems must ensure long-term food and nutrition security, considering availability, access, utilization, and stability [28, 29]. Poudel and Gopinath [30] investigated the variations among global food security indicators provided by organizations like the FAO, the United Nations Development Program, the International Food Policy Research Institute, and the United States Department of Agriculture from 1991 to 2018. However, they focused more on food quantity and nutrition security and paid less attention to climate change security, economic security, social security, and fertilizer security, which can no longer meet the food security assessment requirements under the current development strategy. In other words, ensuring food security requires not only addressing the most fundamental and direct factors but also incorporating a broader range of indirect factors that may impact food security into the research. This comprehensive approach will help refine the food security assessment system over time. Naturally, this assessment system will evolve in response to changes over time and various factors affecting food security, and it is not static. It is a system that gradually aims to provide a more accurate depiction of the evolution of food security.

Additionally, climate change is expected to significantly affect crop, livestock, and fisheries production, and alter the incidence of crop pests [31]. While climate change influences food security, food security also affects climate change [32]. Cauchi, Correa-Velez, and Bambrick [33] reported that climate change is evidently impacting food security and health in Kiribati. A previous study by Kabubo-Mariara and Kenya [34] demonstrated that climate variability and change exacerbate food insecurity, with food security benefiting from favorable agroecological conditions, soil drainage and depth, and high population density. One study found that a decline in the self-sufficiency rate (SSR) can benefit resource management, though it negatively affects food security [35]. Research indicates that reducing food losses in developed countries could decrease the number of undernourished people in developing countries by up to 63 million. This reduction would also lead to decreased harvested land, water usage, and greenhouse gas emissions from food production [36]. In the realm of extreme climate events, another study offered a summary of food security and the effects of climate change, emphasizing the need to incorporate risk reduction strategies into ongoing national and local activities in Bangladesh [37].

In the literature cited above, we reviewed the definition of food security. We also systematically summarized previous studies. Most of these studies have focused on quantitative and nutritional indicators of food security. Few studies have integrated factors, such as food quantity, nutrition, and climate change to assess their individual impacts on food security. This research introduced an indicator system designed to evaluate the role of climate change and other variables in maintaining food security in Japan. A notable feature of this study is its use of quantitative data to illustrate the interactions between food security and climate change, providing readers with a clearer understanding of how climate change affects food security. The food security assessment system includes 16 indicators across six dimensions, covering the four fundamental aspects of food security: availability, access, utilization, and stability of food security. Building upon previous research, this paper also incorporates fertilizer security indicators to assess their long-term impact on food security. The fertilizer security indicators are further subdivided into nitrogen, phosphate, and potash indices. This study analyzed the

indicator system to track changes in food security in Japan from 1980 to 2022 and suggested practical strategies for future improvements in food security.

## 2. Data source and methodology

### 2.1 Evaluation indicator system

Below is an explanation of how the evaluation indicators were chosen. Critical indicators for the food security evaluation system, as outlined in Table 1, were selected based on both the listed literature and additional sources.

Measures such as enhancing food productivity, changing dietary habits, and minimizing food loss and waste can help mitigate the need for land conversion, thereby potentially freeing up land and creating opportunities for the enhanced implementation of other practices, making them essential components of portfolios of practices to address the combined land challenges [38]. Previous research has indicated that adopting a "one health" approach to climate change adaptation may be highly beneficial for maintaining food security [39]. The region's natural resources are under stress from overconsumption, land and marine habitat degradation, freshwater scarcity, and biodiversity loss. Enhanced collaborative efforts, including regional coordination, are crucial for promoting the sustainable use and management of land, forests, water, and marine resources, which are vital for food security and environmental protection [40]. FAO statistics indicate that climate change will significantly affect crop yields, with smallholder agriculture especially susceptible to these risks. Furthermore, Toshiro Nakatsuji [41] examined the effect of changes in the ear emergence period on yield (climatic index),

**Table 1. Food security system for Japan (S1 Data Food security data_Raw data.xlsx).**

| Primary indicators | Secondary indicators | Tertiary indicators | Data sources | Unit |
|---|---|---|---|---|
| Food security (A) | Availability (B1) | Agricultural land (C1)* | FAO | 100 hectares |
| | | Freshwater (C2)* | FAO | % |
| | | Cereal yield (C3)* | FAO | kg |
| | | Over-all grain self-sufficiency (C4)* | World Bank | % |
| | Nutrition (B2) | Undernourishment (C5) | FAO | % |
| | | Prevalence of overweight (C6) | FAO | % |
| | | Total food self-sufficiency (Based on the calorific value of supply) (C7)* | World Bank | % |
| | Society (B3) | Aging (over 65 years old) (C8) | e-Stat | % |
| | Climate change (B4) | CO2 intensity (C9) | World Bank | kg |
| | | Temperature (deviation) (C10) | JMA | ˚C |
| | | Precipitation (deviation) (C11) | JMA | mm |
| | Economics (B5) | Unemployment (C12) | World Bank | % |
| | | Consumer price index (C13) | World Bank | NA |
| | Fertilizers (B6) | Nitrogen (C14) | World Bank | t |
| | | Phosphate (C15) | World Bank | t |
| | | Potash (C16) | World Bank | t |

Note:

* represents a positive indicator; the remainder represent negative indicators.

Data source: FAO, https://www.fao.org/faostat/en/#data/RL

World Bank, https://data.worldbank.org/indicator/EN.GHG.CO2.AG.MT.CE.AR5?locations=JP

e-Stat, https://www.e-stat.go.jp/stat-search/files?page=1&layout=datalist&toukei=00200524&tstat=000000090001&cycle=1&year=20240&month=24101212&tclass1=000001011678

JMA, https://www.data.jma.go.jp/cpdinfo/db/database_temp.html

and it was considered that the decline in yield in response to delayed ear emergence would be slower in the 2030s than in the present, and crop performance would generally be stabilized. Labor is essential to agricultural value chains, from farms to forks [42].

Leon A, Kohyama K, and Yagi K [43] concluded that by accounting for current water conditions using the Tier 2 method, the national estimates of Methane emissions where rice straw was applied could be reduced by 12.7%. Framing economic and social challenges can also be effective in encouraging consumer changes. Unemployment and food prices, to some extent explain the degree of food security [44]. In Japan, the Ministry of Agriculture has promoted various measures to reduce food loss and waste using a combination of different tools, including education, knowledge, science and regulation, and incentive measures. It is difficult to create behavioral changes in consumers and civil society, although appropriate framing in a sociocultural context is potentially effective [45]. Therefore, it is feasible to protect and enhance food security and nutrition while addressing climate change [46]. Further, undernourishment, as well as the prevalence of overweight, affects food security [47]. Besides, the overuse of fertilizers and their inefficient use can equally threaten food security [48]. Greenhouse gases from fertilizer decomposition can lead to environmental pollution [49].

According to previous studies, Japan's overall food security score is 76.5, ranked 21st around the world [42]. The food security score was obtained from the EIU. Although the Ministry of Agriculture, Forestry, and Fisheries has reported that Japan's food supply is secure, its self-sufficiency rate has been approximately 40% for several years [42]. However, when discussing food security research, we find that much of the literature focuses on developing countries, while studies on developed countries are relatively scarce [50]; food security in developed nations is equally important. This study developed a system of food security indicators, detailed in Table 1. Note that missing data are addressed using a weighted average. Table 1 explains each indicator's meanings and data sources.

Table 2 presents the maximum, minimum, mean, and standard deviation for each indicator (S1 Data Food security data_Raw data.xlsx). The range of extremes for freshwater and $CO_2$ intensity is relatively small, and undernourishment shows minimal fluctuation. In contrast, temperature and precipitation, though represented by their variability, still exhibit significant

**Table 2. The descriptive statistics of tertiary indicators (S1 Data Food security data_Raw data.xlsx).**

| Tertiary indicators | Mean | Standard Deviation | Minimum | Maximum |
|---|---|---|---|---|
| Agricultural land | 53310 | 5051.125 | 43250 | 61520 |
| Freshwater | 65.89322 | 1.269621 | 63.29058 | 67.98469 |
| Cereal yield | 5947.488 | 467.5989 | 4429.4 | 6787.3 |
| Over-all grain self-sufficiency | 28.83721 | 2.103628 | 22 | 33 |
| Undernourishment | 2.544186 | 0.138534 | 2.5 | 3.2 |
| Prevalence of overweight | 21.90809 | 4.014669 | 16.2 | 28.93403 |
| Total food self-sufficiency (Based on the calorific value of supply) | 42.86047 | 5.466702 | 37 | 53 |
| Aging (over 65 years old) | 18.26744 | 6.458597 | 9.1 | 28.8 |
| CO2 intensity | 2.471742 | 0.160566 | 2.262539 | 2.77365 |
| Temperature (deviation) | 0.455814 | 0.351271 | 0 | 1.29 |
| Precipitation (deviation) | 143.0744 | 103.5808 | 7.6 | 470 |
| Unemployment | 3.498605 | 1.121527 | 1.95 | 5.57 |
| Consumer price index | 98.14369 | 7.394074 | 77.1626 | 107.8397 |
| Nitrogen | 499776.9 | 117619.2 | 349970.4 | 701000 |
| Phosphate | 533803.9 | 165489 | 309900 | 770000 |
| Potash | 405564.3 | 130244.8 | 209600 | 632400 |

fluctuations. This underscores why temperature and precipitation are considered the primary climate change factors in this study.

## 2.2 Data standardization

Most of the data available for analysis ($y$) were transformed from the original data ($x$) [51]. At the same time, different indicators have different magnitudes and units, and it is important to standardize the indicator data to ensure the scientific nature of the evaluation [52]. Usually, this conversion is standardized in a linear fashion, with the resulting value ranging from 0 to 1. The formula for standardization is as follows:

$$y_{ij} = \frac{x_{ij} - minx_i}{maxx_i - minx_i} (Positive) \tag{1}$$

$$y_{ij} = \frac{maxx_i - x_{ij}}{maxx_i - minx_i} (Negative) \tag{2}$$

Here, $x_{ij}$ denotes the raw value of the $j$-th indicator in the $i$-th year, while $y_{ij}$ represents the standardized index value. Note that in Eqs (1) and (2), the terms "positive" and "negative" within the parentheses denote the standardization methods for positive and negative indicators, respectively (S2 Data Food security data_Data standardization.xlsx).

## 2.3 Coefficient of variance

The coefficient of variation is a key metric for assessing the stability of an indicator; a lower coefficient signifies greater stability [53]. Indicator weights reflect the significance of each indicator within the evaluation system. According to the method used to calculate these weights, a higher coefficient of variation results in greater indicator weights. Notably, the importance of the indicators mentioned here is not absolute, but rather reflects the relative significance between the indicators. The indicators we set in our evaluation indicator system are all important indicators of food security.

Establishing indicator weights is crucial for developing an effective indicator system. Currently, weight assignment methods are generally categorized into subjective and objective approaches, including techniques such as the Delphi method and Analytic Hierarchy Process (APH); the latter mainly includes the deviation maximization, entropy value, and coefficient of variation methods [54]. Subjective assignment methods are easily affected by human factors. Consequently, this study utilized the coefficient of variation method to assign weights to each index. The steps for calculating the coefficient of variation are outlined below.

First, the average value of each indicator was calculated. $Y_j$ stands by the mean of $j$-th indicator, $m \in Z^+$.

$$Y_j = \frac{1}{n} \sum_{i=1}^{} y_{ij}, j = 1, 2 \ldots, m \tag{3}$$

Next, the standard deviation for each indicator was determined, with $S_j$ representing the standard deviation of $j$-th indicator.

$$S_j = \sqrt{\frac{1}{n-1} \sum_{i=1}^{n} (y_{ij} - Y_j)^2}, j = 1, 2, \ldots, m \tag{4}$$

Finally, the CVs of all indicators can be calculated using Eq (4). The coefficient of variation is the ratio of the mean to the standard deviation, and this method provides an objective way

to calculate weights by directly analyzing the data for f each indicator, which can effectively and objectively reflect the gaps in each indicator [55]. $V_j$ indicates the coefficient of variation of $j$-th indicator.

$$V_j = \frac{S_j}{Y_j}, j = 1, 2, \ldots, m \tag{5}$$

The weight of each indicator is the CV of a single indicator divided by the sum of the CVs of all indicators. $W_j$ is the weight of the $j$-th indicator.

$$W_j = \frac{V_j}{\sum_{j=1}^{m} V_j}, j = 1, 2, \ldots, m \tag{6}$$

Level 2 indicator values can be obtained by adding the normalized level 3 indicator values according to the linear weighting method. $F_i$ represents the weight of $j$-th indicator in $i$-th year, $n \in Z^+$.

$$F_i = \sum_{j=1}^{n} y_{ij} W_j, i = 1, 2, \ldots, n \tag{7}$$

It is noteworthy that the calculation method for the composite index of primary indicators (commonly referred to as the food security composite index) is similar to the method used for assigning weights to tertiary indicators [56].

## 3. Results and discussion

### 3.1 Food security in Japan

Using the food security evaluation index system and weights described above, the food security index for Japan and the six subsystem security indices (S3 Data Food security data_Level 2 & 1 indicators weight.xlsx) were calculated for the period from 1980 to 2022. The results are displayed in Table 3 (S4 Data Food security data_Level 3 indicators weight.xlsx) and Fig 1 (S5 Data Food security data_Figure 1.xlsx).

Indicators that have a "positive" impact on food security imply that higher values of the indicator lead to an increase in the food security index, while "negative" indicators have the opposite effect. The final column displays the weights assigned to each level of the indicators. The weight for a secondary indicator is derived from the sum of the weights of the corresponding tertiary indicators within each group. The score for a primary indicator is calculated based on data from the secondary indicators, following a similar process as used for the tertiary indicators. According to the weighting results in Table 3, the tertiary indicators are ranked by importance as follows: total food self-sufficiency (based on the calorific value of supply), consumer price index, phosphate, aging (over 65 years old), nitrogen, potash, prevalence of overweight, unemployment, $CO_2$ intensity, agricultural land, freshwater, temperature (deviation), precipitation (deviation), cereal yield, over-all grain self-sufficiency, and undernourishment.

According to the line of food security index, Japan's food security is expected to decline from 1980 to 2022. Food security declined steeply in 1993. This was because Japan experienced a rice crisis in 1993 due to a bad harvest since experienced cold weather [57]. We observed that the Climate Security Index exhibited greater volatility compared to other subsystem indices, which is closely related to Japan's highly variable climate. Rice cultivation, which spans from April to October each year, coincides with the peak period for extreme weather events. According to Japan's Climate Yearbook and meteorological statistics, extreme weather events between 1980 and 2022 included the following: Climate instability marked by heavy rainfall,

**Table 3. Weights for tertiary indicators (S4 Data Food security data_Level 3 indicators weight.xlsx).**

| Primary indicators | Secondary indicators | Tertiary indicators | Weight (Unit: 100%) |
|---|---|---|---|
| Food security (A) | Availability (B1) | Agricultural land (C1)* | 0.060007 |
| | | Freshwater (C2)* | 0.0583 |
| | | Cereal yield (C3)* | 0.036812 |
| | | Over-all grain self-sufficiency (C4)* | 0.03677 |
| | Nutrition (B2) | Undernourishment (C5) | 0.025245 |
| | | Prevalence of overweight (C6) | 0.06829 |
| | | Total food self-sufficiency (Based on calorific value of supply) (C7)* | 0.111481 |
| | Society (B3) | Aging (over 65 years old) (C8) | 0.073285 |
| | Climate change (B4) | CO2 intensity (C9) | 0.063561 |
| | | Temperature (deviation) (C10) | 0.050326 |
| | | Precipitation (deviation) (C11) | 0.037865 |
| | Economics (B5) | Unemployment (C12) | 0.064708 |
| | | Consumer price index (C13) | 0.091138 |
| | Fertilizers (B6) | Nitrogen (C14) | 0.069857 |
| | | Phosphate (C15) | 0.083735 |
| | | Potash (C16) | 0.068621 |

Note:

* represents a positive indicator; the remainder represent negative indicators. All indicator weights were calculated using the coefficient of variation method and summed to one.

typhoon-prone years, frequent flooding, droughts in Okinawa, and heavy rain in Nagasaki, leading to three consecutive years of poor rice harvests during 1980–1982. Consecutive years of extreme summer heat and low rainfall during 1983–1986. Multiple typhoons, heavy rainfall, and heat waves in 1989. Generally high annual temperatures, frequent typhoons, and reduced

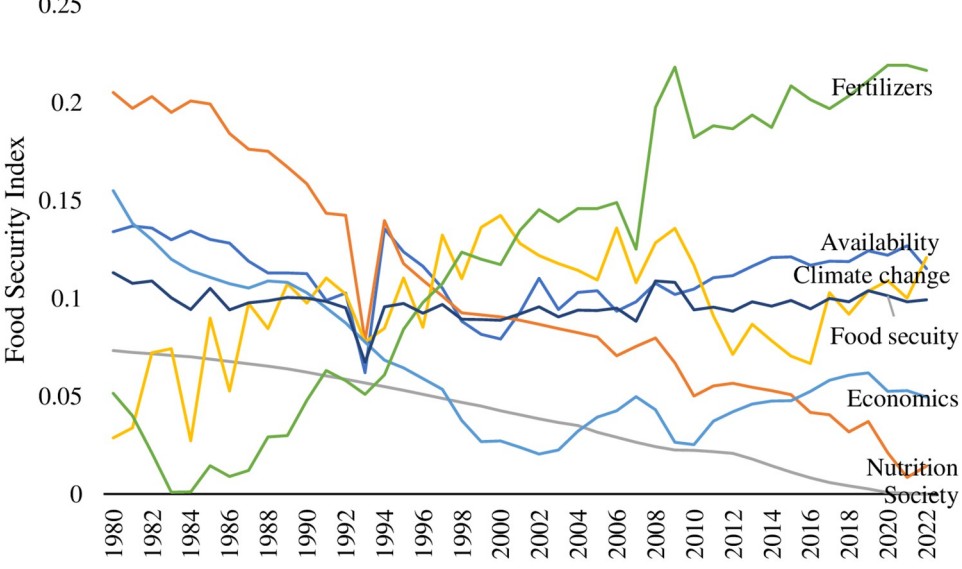

**Fig 1. Food security development in Japan, 1980–2022 (S5 Data Food security data_Figure 1.xlsx).**

sunlight duration during 1998–2000. Similar patterns of summer heat, typhoons, heavy rainfall, and reduced sunlight duration during 2012–2016. These climatic fluctuations align with the findings of this study.

This part presents the weights of the six subsystems. Note that each subsystem's weight is the sum of the weights of its respective indicators. As shown in Table 3, among the six subsystems of food security, the food fertilizer security subsystem had the largest weight at 0.222213, followed by food nutritional security at 0.205116, food availability security ranks third at 0.191889, followed by food economic security at 0.155846. The climate change security is ranked fifth with a score of 0.151752, and food social security ranks last with a score of 0.073285. This approach differs significantly from international studies on food security, which typically emphasize the four dimensions of food availability, access, utilization, and stability.

The study categorizes Japan's food security into three stages from 1980 to 2022. The first stage is a decline from 1980 to 1993. The second stage is the general rising stage, from 1994 to 2009. The third stage is the stable development stage, from 2010 to 2022. As shown in Fig 1, the food security index was high between 2008 and 2009. This is supported by the Fertilizer Security Index and Climate Change Security Index.

The Climate Change Security Index (CCSI) displayed an overall upward trend from 1980 to 2000. However, from 2000 to 2016, the index generally declined. Following 2016, the index began to rise once more. These three periods showed relatively large fluctuations in climate security indicators. Climate change is also a factor of uncertainty [31, 58, 59]. Based on the definition of the coefficient of variation mentioned in 2.3 and the CV for carbon dioxide density, temperature, and rainfall, we found that carbon dioxide density is the most significant factor affecting the Climate Change Security Index. Variations in atmospheric carbon dioxide levels and carbon dissolved in water can lead to regional warming or cooling [60]. Human-induced climate change impacts both the quality and quantity of food production and its equitable distribution [13]. However, while climate change mitigation aimed solely at meeting climate goals are essential, they may inadvertently pose risks to food security. The effectiveness of mitigation policies can significantly influence how well people at risk of hunger respond to these measures [61]. Therefore, people should be proactive in addressing potential food insecurity associated with climate change.

The food availability security index showed a declining trend from 1980 to 1993. After 1993, it showed a fluctuating growth trend. On one hand, this contributes positively to the overall food security index; however, it may also somewhat undermine the nutrition, social, climate change, and economic aspects of food security. Maintaining sufficient agricultural land is crucial for sustaining the country's food security. Due to the social factors of aging and the decreasing number of agricultural workers, agricultural land in Japan is diminishing year by year. Consequently, Japan's annual food production is steadily declining, accompanied by a simultaneous decrease in the country's food self-sufficiency rate. To a certain extent, this has affected the food self-sufficiency rate, which, in turn, affects food security. Output per unit of cereal is also a key factor in ensuring food security. It is more severely affected by climate change; for example, in 1993, the unit output was the lowest between 1980 and 2022.

Nutrition and social security indices showed a declining trend from 1980 to 2022. The decline in these two indices hindered the development of food security in Japan. From 1980 to 2022, the overall undernourishment rate in Japan did not exceed 2.5%. This provides support for food nutrition security. Nonetheless, the prevalence of overweight individuals is steadily rising each year, which poses a risk to food nutrition security. Fig 1 shows that the weight of the total food self-sufficiency (calories) is 11.1481%. This indicates that the total food self-sufficiency (calories) rate is high in the food security evaluation system. As Japan's total food

self-sufficiency (calories) rate is decreasing annually, it needs to be taken seriously. Although there are studies that show that Japan is nutritionally balanced [62, 63], there are also studies that show that nutritional security is quite important for ensuring food security [64]. The findings of this study indicate that the nutrition security component of the food security system is deteriorating. This in a way shows that nutrition security in Japan is a concern, for example, nutrition is not ensured in poor areas of Japan [65]. Japan's aging population was projected to increase from 6.3% in 1980 to 28.8% in 2022, reflecting an average annual growth rate of 22.7%. The intensifying aging population has led to a shortage of agricultural workers in Japan, directly affecting food production and subsequently lowering the food security index. As shown in Fig 1, aging is an impediment to food security in Japan.

The Economic Security Index (ESI) shows a downward trend from 1980 to 2003 and an overall upward trend from 2003 to 2022. Overall, food economic security provides strong support for food security during 2003–2022.

The Fertilizer Security Index showed a general increase from 1980 to 2022, providing strong support for food security. It is evident that incorporating fertilizer security indicators into the food security indicator system is highly necessary, as this provides valuable reference for future assessments of food security. The food nutrition security subsystem exhibited an overall rapid downward trend. Food nutrition security is a hidden problem for Japan's food security.

## 3.2 Discussion

According to the constructed evaluation index system, the food fertilizer security subsystem is the primary factor affecting the food security system. Overall, there has been a rapid upward trend from 1980 to 2022, which significantly supports food security in Japan. Unlike the food fertilizer security system, the food climate change security system negatively affected the food security index during the periods from 1980 to 1990 and from 1997 to 2009. Additionally, from 2012 to 2017 and 2013 to 2022, it played a negative and supporting role, respectively. This indicates that the impact of environmental changes on food security is characterized by significant complexity and uncertainty [66]. Both the food supply security subsystem and the food climate change security subsystem exhibit dual effects on food security, meaning they can have both positive and negative impacts. However, the development of the food availability security subsystem was relatively stable. This indicates that Japan's development in terms of availability is relatively stable. Additionally, the food and social security subsystem was consistently below the food security index. This indicates that Japan's aging population has always been a hidden threat to food security [67].

This study investigated the variations in the food climate change security subsystem and their effects on food security. From the standpoint of sudden changes in this subsystem, its influence on food security has evolved over time. In other words, climate change has been a hidden threat to ensuring food security from 1980 to 2022. Climate change cannot be fundamentally stabilized. However, lessons from historical climate extremes can guide efforts to better prepare for potential future extremes. Consequently, proactively addressing climate change is crucial for maintaining a robust climate change security index for food. Other subsystems that affect the food security index should also maintain stable growth to support Japan's food security. Sri Lanka is experiencing food security issues as a result of climate change, with researchers suggesting strategies to enhance food sovereignty as a way to alleviate these effects [68]. In Africa, where the impact of global warming is particularly severe, experts have advised focusing on land management and other measures to address the challenges [69]. The emergence of Japan Society 5.0 has established a new form of disaster prevention as well as climate change [70].

Nevertheless, since climate change is a long process [71], we have only studied one period of it, and the impact of internal climate change on food security in other periods remains to be studied. We did not examine the long-term effects of climate change on food security due to a lack of data on various impact factors. Given the complexity of factors influencing food security, we could only address some of them due to data availability constraints.

## 4. Conclusions and future work

The study carried out an extensive evaluation of Japan's food security and six related indicators by developing the Japan Food Security Indicator System. The results showed that the climate change security indicator experienced the most seismic changes. Climate change has a profound impact on Japan's food security. The relationship between the Climate Change Security Index and the Food Security Index reveals an inverted U-shaped pattern. Specifically, the influence was negative from 1980 to 1989, positive from 1990 to 2011 (with exceptions in 1993 and 1997), and negative again from 2012 to 2016. This pattern illustrates that the effects of climate change on food security are not linear; instead, they fluctuate based on the severity of climate change and the capacity for adaptation. Consequently, effectively managing climate change requires a holistic approach that takes into account both the impacts of climate change and the necessary adaptation strategies to safeguard food security. With global warming, Japan's climate has undergone corresponding changes, such as an increase in extreme weather events and an uneven distribution of precipitation. These changes, including interruptions in crop growth cycles and increased pest and disease activity, have significantly affected agricultural production in Japan [72]. From 1980 to 2022, Japan's food security faced many challenges. Climate change-induced decreases in agricultural production and declines in the quality of agricultural products have increased Japan's vulnerability regarding food security [73]. In addition, external factors such as fluctuations in the global food market and geopolitical risks have affected Japan's food security to some extent [74]. Japan has adopted various strategies to address the challenges of climate change. The government has developed policies to advance agricultural science and technology, enhance production efficiency, and develop pest-resistant crop varieties [70, 75, 76]. Simultaneously, Japan's agricultural industry chain is constantly adjusting to the impacts of climate change [77]. Enterprises and individuals have invested in areas such as food production enhancement technology and smart agriculture to turn the crisis into an opportunity. While Japan has made progress in mitigating the effects of climate change on food security, it will face increasing challenges in the future. As climate change escalates, agricultural production in Japan is likely to encounter greater uncertainty, keeping food security issues at the forefront. According to a report from the Ministry of Agriculture, Forestry, and Fisheries of Japan, global warming is expected to have a significant impact on Japanese agriculture. The report states that, by about 2060, crop failures and other phenomena caused by "high temperatures" in the zone south of the north-east of Japan will reduce rice production in Japan, while in the west of the country there will be a significant reduction in rice production [78]. Therefore, it is crucial for Japan to continue advancing agricultural science and technology, promote industrial restructuring, and enhance international cooperation to effectively address climate change and ensure food security. Additionally, annual adjustments in international trade can be vital in redirecting food supplies from surplus regions to areas experiencing shortages due to extreme weather events, civil strife, and/or other disruptions [79]. Farmers can also adapt to new climatic conditions by switching to completely different crops or redistributing the land from crop production to grazing [13]. As a member of FAO, Japan plays a significant role in promoting food security. Japan's efforts to tackle climate change will also serve as a model for other member countries and nations dealing with the effects of climate

change. In addition, we have made an unexpected new finding: the support provided by the Fertilizer Security Index to the Food Security Index is quite significant. This also indicates that indirect factors affecting food security are important, cannot be ignored, and have consistently been present.

This study specifically focuses on climate change. However, the results indicate that, in addition to climate change, the impact of the fertilizer security index on food security should not be underestimated. Simultaneously, the indispensable factors for evaluating the system of indicators have been added to the article. However, certain factors were not fully considered. Given the complexity of the factors influencing food security, variables such as the prevalence of food pests and diseases, pesticide residues, and water quality should be incorporated into food security assessment systems. However, due to the lack of available data on these aspects, we are currently unable to evaluate them. Therefore, we will upgrade this food security evaluation system in the future to achieve a more accurate assessment of long-term changes and interactions between climate change and food security indicators.

## Supporting information

**S1 Table. Food security system for Japan.** This table provides the food security indicator system and the data sources for each indicator.
(PDF)

**S2 Table. The descriptive statistics of tertiary indicators.** The range of extremes for freshwater and $CO_2$ intensity is relatively small, and undernourishment shows minimal fluctuation. In contrast, temperature and precipitation, though represented by their variability, still exhibit significant fluctuations. This underscores why temperature and precipitation are considered the primary climate change factors in this study.
(PDF)

**S3 Table. Weights for Tertiary indicators.** This table lists the weights of each tertiary indicator within the food security indicator system. Among them, *Total food self-sufficiency (based on the calorific value of supply)* has the highest weight, indicating that it is the most critical indicator for ensuring food security within the study period and scope.
(PDF)

**S1 Text.** The formula for standardization is as follows:

$$y_{ij} = \frac{x_{ij} - minx_i}{maxx_i - minx_i} \, (Positive) \tag{1}$$

$$y_{ij} = \frac{maxx_i - x_{ij}}{maxx_i - minx_i} \, (Negative) \tag{2}$$

Here, $x_{ij}$ denotes the raw value of the *j*-th indicator in the *i*-th year, while $y_{ij}$ represents the standardized index value. Note that in Eqs (1) and (2), the terms "positive" and "negative" within the parentheses denote the standardization methods for positive and negative indicators, respectively.
This section provides a detailed explanation of the principles behind the data normalization used in this study.
(PDF)

**S1 Fig. Food security development in Japan, 1980–2022.** The figure categorizes Japan's food security into three stages from 1980 to 2022. The first stage is a decline from 1980 to 1993. The

second stage is the general rising stage, from 1994 to 2009. The third stage is the stable development stage, from 2010 to 2022. The food security index was high between 2008 and 2009. This is supported by the Fertilizer Security Index and Climate Change Security Index.
(PDF)

**S1 Data.**
(XLSX)

**S2 Data.**
(XLSX)

**S3 Data.**
(XLSX)

**S4 Data.**
(XLSX)

**S5 Data.**
(XLSX)

## Acknowledgments

We thank Prof. Ken'ichi Matsumoto, Toyo University, for his valuable suggestions. Also, we would like to thank Editage (www.editage.cn) for English language editing.

## Author Contributions

**Conceptualization:** Caixia Li.

**Data curation:** Caixia Li.

**Formal analysis:** Caixia Li.

**Methodology:** Caixia Li.

**Resources:** Caixia Li.

**Software:** Caixia Li.

**Supervision:** Caixia Li.

**Validation:** Caixia Li.

**Visualization:** Caixia Li.

**Writing – original draft:** Caixia Li.

**Writing – review & editing:** Caixia Li.

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
