## [Decision Letter · Decision Letter 0]

4 Jul 2024

PONE-D-24-11471The evolution of food security in Japan --Based on an indicator evaluation system including climate change indicatorsPLOS ONE

Dear Dr. LI,

Thank you for submitting your manuscript to PLOS ONE. After careful consideration, we feel that it has merit but does not fully meet PLOS ONE’s publication criteria as it currently stands. Therefore, we invite you to submit a revised version of the manuscript that addresses the points raised during the review process.

Please see the reviewer comments and make all correction carefully pointwise.==============================

We look forward to receiving your revised manuscript.

Kind regards,

Pradeep Mishra

Academic Editor

PLOS ONE

Journal Requirements:

Reviewers' comments:

Reviewer's Responses to Questions

**Comments to the Author**

1. Is the manuscript technically sound, and do the data support the conclusions?

Reviewer #1: Partly

Reviewer #2: Yes

2. Has the statistical analysis been performed appropriately and rigorously? 

Reviewer #1: No

Reviewer #2: Yes

3. Have the authors made all data underlying the findings in their manuscript fully available?

Reviewer #1: No

Reviewer #2: No

4. Is the manuscript presented in an intelligible fashion and written in standard English?

Reviewer #1: No

Reviewer #2: Yes

5. Review Comments to the Author

Reviewer #1: The study provides a comprehensive assessment of Japan's food security and its relationship with various indicators, particularly focusing on the impact of climate change.

Abstract:

It could be improved by including specific quantitative results or trends observed in Japan's food security index evolution. Additionally, it would be helpful to mention the novelty or contribution of the study to the existing literature on food security and climate change.

Introduction:

• The introduction could be more focused by briefly stating the research objectives and hypotheses guiding the study.

• Need to critically evaluate and integrate the existing literature to identify gaps or limitations that the current study aims to address.

Methodology

• It would be beneficial to provide more explicit justification for the inclusion/exclusion of certain indicators, especially considering the complexity and multidimensionality of food security.

• Provide a brief rationale for why data standardization was necessary and how it contributes to the analysis.

• Include a discussion of potential limitations or assumptions underlying the methodology, such as data reliability or sensitivity to parameter choices.

Results and Discussion

• Specify the factors contributing to the fluctuations in the Climate Change Security Index (CCSI) to provide a clearer understanding of the dynamics.

• Explain the significance of the decline in the food nutrition security subsystem and its potential implications for food security in Japan.

Conclusion

• State the key findings of the study, emphasizing the impact of climate change on Japan's food security.

• Mention the concrete examples of how climate change has affected agricultural production in Japan to illustrate the severity of the issue.

• Emphasize the significance of Japan's current measures to address climate change and ensure food security, such as agricultural innovation and international cooperation.

Reviewer #2: Introduction

The introduction effectively sets the stage by emphasizing the importance of food security, particularly in the context of climate change. The authors highlight Japan's role in global food security initiatives and the challenges it faces due to climate change and other factors. The literature review is thorough, summarizing various studies related to food security and climate change, and establishing the need for a comprehensive evaluation system that includes climate change indicators.

Data and Methods

The authors should provide comprehensive details about the data they used. The methodology section is detailed and well-structured. The authors describe the selection of evaluation indicators comprehensively. The six dimensions of the food security evaluation system (availability, nutrition, climate change, society, economics, and fertilizer) are clearly defined. The use of the coefficient of variation method to determine indicator weights is appropriate and well-explained. However, some parts could benefit from more clarity, especially the calculation steps for the coefficient of variation and the integration of Level 1, 2, and 3 indicators. On page 13, equation (3) should define 'm'. Likewise, equation (4) should define 'n'.

Results

The results section provides a detailed analysis of Japan's food security index from 1980 to 2022. The categorization into three stages (1980-1993, 1994-2009, and 2010-2022) helps in understanding the trends and fluctuations in food security. The discussion on the impact of different subsystems (fertilizer, climate change, availability, etc.) on food security is insightful. The visual representation through tables and figures aids in comprehending the trends, though the clarity and readability of the figures could be improved. On page 14, figure one should be on line 251.

Discussion

The discussion effectively interprets the results, linking them back to the literature reviewed. The authors successfully highlight the complexity and variability of the impact of climate change on food security. The emphasis on the need for proactive measures to address potential food insecurity associated with climate change is well-placed. However, the discussion could be enhanced by integrating more recent studies and providing a comparative analysis with other countries facing similar issues.

Conclusions

The conclusions summarize the key findings and emphasize the significant impact of climate change on Japan's food security. The recommendations for future actions, such as strengthening agricultural scientific and technological innovations and promoting international cooperation, are relevant and actionable. The acknowledgment of limitations and the call for further research to upgrade the evaluation system is appropriate.

Grammatical and Stylistic Review

The paper is generally well-written, but it does have some grammatical errors and stylistic issues that need attention. Here are a few examples:

1. Grammatical Errors:

"Owing to the large fluctuations in climate change, the impact of climate change security indicators on food security cannot be ignored." (needs rephrasing for clarity). "The study categorized Japan’s food security into three stages from 1980 to 2022." (tense consistency)

2. Stylistic Issues:

Repetition of phrases like "food security" could be minimized for better readability. Some sentences are overly complex and could be simplified for clarity.

Recommendations for Improvement

1. Clarify Methodology: Provide more detailed steps for the coefficient of variation calculation and data standardization processes.

2. Enhance Figures and Tables: Improve the readability and clarity of figures and tables.

3. Integrate Recent Studies: Include more recent studies in the discussion section for a comprehensive analysis.

4. Grammar and Style: Conduct a thorough grammatical review and simplify complex sentences for better readability.

Conclusion

The paper provides a comprehensive evaluation of Japan’s food security, integrating climate change indicators into the assessment. While the methodology is robust and the findings are insightful, improvements in clarity, recent literature integration, and grammatical accuracy will enhance the overall quality and impact of the paper.

6. PLOS authors have the option to publish the peer review history of their article (what does this mean?). If published, this will include your full peer review and any attached files.

Reviewer #1: **Yes: **Professor Shankarappa Sridhara

Reviewer #2: No

---

## [Author Response · Author response to Decision Letter 0]

28 Jul 2024

Dear Reviewers and Editors,

Thanks to the reviewers and editors for the suggestions. We have revised and improved the manuscript based on your suggestions. Thank you again for your valuable suggestions.

Thanks & best regards,

CAIXIA LI

---

## [Decision Letter · Decision Letter 1]

4 Sep 2024

PONE-D-24-11471R1The evolution of food security in Japan

--Based on an indicator evaluation system including climate change indicatorsPLOS ONE

Dear Dr. LI,

Thank you for submitting your manuscript to PLOS ONE. After careful consideration, we feel that it has merit but does not fully meet PLOS ONE’s publication criteria as it currently stands. Therefore, we invite you to submit a revised version of the manuscript that addresses the points raised during the review process.

We look forward to receiving your revised manuscript.

Kind regards,

Pardeep Singh

Academic Editor

PLOS ONE

Journal Requirements:

Reviewers' comments:

Reviewer's Responses to Questions

**Comments to the Author**

1. If the authors have adequately addressed your comments raised in a previous round of review and you feel that this manuscript is now acceptable for publication, you may indicate that here to bypass the “Comments to the Author” section, enter your conflict of interest statement in the “Confidential to Editor” section, and submit your "Accept" recommendation.

Reviewer #1: All comments have been addressed

Reviewer #2: All comments have been addressed

2. Is the manuscript technically sound, and do the data support the conclusions?

Reviewer #1: Partly

Reviewer #2: Yes

3. Has the statistical analysis been performed appropriately and rigorously? 

Reviewer #1: Yes

Reviewer #2: Yes

4. Have the authors made all data underlying the findings in their manuscript fully available?

Reviewer #1: Yes

Reviewer #2: Yes

5. Is the manuscript presented in an intelligible fashion and written in standard English?

Reviewer #1: Yes

Reviewer #2: Yes

6. Review Comments to the Author

Reviewer #1: Abstract: The inclusion of specific quantitative results and trends in the revised abstract is a positive addition. However, the highlighted innovations, such as the introduction of the Fertilizer Security Index (FSI), could be more explicitly connected to how they advance the understanding of food security in the context of climate change. Clarifying the novelty of this index in comparison to existing studies would strengthen the abstract further.

Introduction: The authors have improved the introduction by including the research objectives and hypotheses. The revisions provide a clearer focus on the study's goals.

Methodology: The revisions have adequately addressed the need for justification of the inclusion/exclusion of indicators. The explanation of data standardization is clearer, which enhances the understanding of its importance to the analysis.

Results and Discussion: The authors' addition of the factors contributing to fluctuations in the Climate Change Security Index (CCSI) is a valuable enhancement. The discussion on the decline in the food nutrition security subsystem is also more detailed, which helps clarify its significance.

Overall, the authors have made significant improvements in response to my comments. I believe the manuscript is now much stronger and may be accepted for the publication.

Reviewer #2: (No Response)

7. PLOS authors have the option to publish the peer review history of their article (what does this mean?). If published, this will include your full peer review and any attached files.

Reviewer #1: No

Reviewer #2: No

---

## [Author Response · Author response to Decision Letter 1]

8 Sep 2024

Thank you for all kind of comments.

---

## [Decision Letter · Decision Letter 2]

17 Dec 2024

PONE-D-24-11471R2The evolution of food security in Japan

--Based on an indicator evaluation system including climate change indicatorsPLOS ONE

Dear Dr. LI,

Thank you for submitting your manuscript to PLOS ONE. After careful consideration, we feel that it has merit but does not fully meet PLOS ONE’s publication criteria as it currently stands. Therefore, we invite you to submit a revised version of the manuscript that addresses the points raised during the review process.

**ACADEMIC EDITOR: **Critical concerns and comments have been raised by one of the reviewers. After discussion with other Editors, I  maintained that the authors had made substantial revisions to the manuscript and had responded to the previously raised comments. However, the comments below are important and should be addressed as well.

We look forward to receiving your revised manuscript.

Kind regards,

Edwin Hlangwani

Academic Editor

PLOS ONE

Journal Requirements:

Reviewers' comments:

Reviewer's Responses to Questions

**Comments to the Author**

1. If the authors have adequately addressed your comments raised in a previous round of review and you feel that this manuscript is now acceptable for publication, you may indicate that here to bypass the “Comments to the Author” section, enter your conflict of interest statement in the “Confidential to Editor” section, and submit your "Accept" recommendation.

Reviewer #3: All comments have been addressed

Reviewer #4: (No Response)

2. Is the manuscript technically sound, and do the data support the conclusions?

Reviewer #3: Yes

Reviewer #4: No

3. Has the statistical analysis been performed appropriately and rigorously? 

Reviewer #3: Yes

Reviewer #4: I Don't Know

4. Have the authors made all data underlying the findings in their manuscript fully available?

Reviewer #3: Yes

Reviewer #4: No

5. Is the manuscript presented in an intelligible fashion and written in standard English?

Reviewer #3: Yes

Reviewer #4: No

6. Review Comments to the Author

Reviewer #3: (No Response)

Reviewer #4: 1. Introduction

“56-58 The research hypothesis suggests that the Climate Change Security Index’s influence on food security fluctuated between 1980 and 2022.”

The above hypothesis is the actual research objective of the author. The many generalities about climate change in the introduction and literature review are not focused and often poorly stated or reviewed.

The author should focus on a detailed, comprehensive description and literature review of the six subsystem security indices, which constitute the content and methodology of the conducted research.

The reader gets quickly lost in the current organization and writing of the manuscript about the actual purpose and objective of the research.

2. Data source and methodology

An example of unnecessary generalities and erroneous statements:

“212-214 Factors contributing to climate change include rising temperatures, rising sea levels, and excess rainfall.”

This is a non-sense statement, because these are not factors contributing to climate change, but may be the results of climate change. Ironically, the only example the author gives about climate change impact in Japan is a lowering of temperature, as cold weather negatively impacted the rice yields in 1969:

This is mentioned 3 times in the manuscript:

“20-21 In 1993, a rice crisis occurred owing to cold weather.

357-358 Food security declined steeply in 1993. This was because Japan experienced a rice crisis in 1993 due to a bad harvest caused since experienced cold weather.

501-502 Japan was hit by an extreme cold weather in 1993, which seriously threatened food security.”

The actual sources of the data used in the research are not presented in a transparent manner. Also the methodology used is difficult to comprehend for the reader. Seven mathematical formulas are given as a kind of black box. But how the data are put into these formulas, and which data are put in, and how the results come out of the black box is not clear from the manuscript.

“256-259 This study developed a system of food security indicators, detailed in Table 1.”

The data sources mentioned for each item in Table 1 only mention the respective organizations (FAO, World Bank, e-Stat, JMA) but without giving a clear reference (author, year, journal or web site address) to the actual publication in which the data can be found. Moreover, the data given in last column in Table 1, i.e. Weights (100%), is confusing for the reader. The question arises whether these data in the last column constitute a data source or are the results of the author’s research???

“265-266 The final column displays the weights assigned to each level of the indicators.”

This statement by the author in lines 265-266 seems to suggest that these are apparently the principal results of the author’s study. This is highly confusing, as these data are presented in the methodology section and not in the section about the results.

3. Results and discussion

“349-353 Using the food security evaluation index system and weights described above, the food security index for Japan and the six subsystem security indices were calculated for the period from 1980 to 2018. The results are displayed in Table 1 and Figure 1.”

However, no detailed descriptions or examples are given how the mathematical formulas produced the research results in Table 1 and Figure 1.

What is also missing in the text is a clear reference to the supporting data information, not included in the article, which forms an impressive list of data for the period 1980-2018.

The text does not give a clear explanatory link between the origin of these data (author, year, journal or web site address), their actual input for calculation into the 7 mathematical formulas, and how this produced the research results.

The supporting data information excel file is neither transparent nor understandable for the reader.

The author should perhaps make an individual table for each of the six investigated subsystem security indices for the years 1980-2018, including detailed explanations in the text of the manuscript.

The structure of the article ought to be completely changed and focused on the actual research carried out by the author and not on generalities about climate change.

Also the introduction must be focused on Japan, including a literature review of climate changes or extreme weather events that hit Japan in the past century in relation to food security, as Japan is the subject of the investigation.

In its present form the manuscript is quite incomprehensible and not suited for publication.

7. PLOS authors have the option to publish the peer review history of their article (what does this mean?). If published, this will include your full peer review and any attached files.

Reviewer #3: **Yes: **Ghassan Abdul-Majeed

Reviewer #4: No

---

## [Author Response · Author response to Decision Letter 2]

22 Dec 2024

Thank you for all your comments and suggestions on the manuscript. We have provided responses to each of your remarks and recommendations. If there are any further revisions needed in the manuscript, we would be happy to make the necessary improvements. Once again, we sincerely appreciate your feedback, which has greatly improved the quality of the manuscript.

---

## [Editor Report · Decision Letter 3]

23 Dec 2024

The evolution of food security in Japan

--Based on an indicator evaluation system including climate change indicators

PONE-D-24-11471R3

Dear Dr. LI,

We’re pleased to inform you that your manuscript has been judged scientifically suitable for publication and will be formally accepted for publication once it meets all outstanding technical requirements.

Kind regards,

Edwin Hlangwani

Academic Editor

PLOS ONE
---

## [Editor Report · Acceptance letter]

9 Jan 2025

PONE-D-24-11471R3 

PLOS ONE

Dear Dr. LI, 

I'm pleased to inform you that your manuscript has been deemed suitable for publication in PLOS ONE. Congratulations! Your manuscript is now being handed over to our production team.

Kind regards, 

on behalf of

Dr. Edwin Hlangwani 

Academic Editor

PLOS ONE